# Variation in Soil Aggregate Stability Due to Land Use Changes from Alpine Grassland in a High-Altitude Watershed

Ying Li [1], Zhanming Ma [1], Yutao Liu [1], Zilong Cui [1], Qiuyu Mo [1], Can Zhang [1], Haiyan Sheng [2], Wen Wang [1] and Yongkun Zhang [1,*]

[1] State Key Laboratory of Plateau Ecology and Agriculture, Qinghai University, Xining 810016, China
[2] College of Agriculture and Animal Husbandry, Qinghai University, Xining 810016, China
* Correspondence: zhangyongkun321@163.com

**Abstract:** Land use change affects soil aggregate composition and stability, which impacts soil structure and health. To reveal how land use change impacted soil aggregates of alpine grassland in a high-altitude watershed, soil samples from 161 sites including alpine grassland, cropland and abandoned land were selected to measure and analyze the distribution of aggregate fractions (macro-aggregates, micro-aggregates, silt+clay), soil aggregate stability (mean weight diameter, geometric mean diameter, fractal dimension, etc.) and related soil properties (soil organic carbon content, soil particle composition, etc.) in the Huangshui River watershed of the Qinghai–Tibet Plateau. The results showed: (1) As alpine grasslands were converted to croplands and croplands to abandoned lands, the proportion of macro-aggregates and the aggregate stability index showed a trend of first decreasing and then increasing ($p < 0.05$), indicating that tillage and abandonment have significant influences on soil aggregate structure. (2) Compared with temperate grassland, alpine grassland had richer soil organic carbon, and a higher ratio of macro-aggregates and aggregate stability. (3) Soil organic carbon and sand content had distinct influences on the fractions and stability of aggregates during land use change. These results suggested that cultivation can substantially reduce the soil aggregate stability in alpine grassland, whereas abandonment can effectively improve soil aggregate structure.

**Keywords:** cropland; abandoned land; soil particle composition; soil structure; soil organic carbon

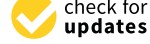



## 1. Introduction

Soil aggregates are the fundamental structural units that govern the dynamics of soil organic matter (SOM) and nutrient cycling [1,2]. Aggregate stability is a vital index to measure soil structure and physical shape, while evaluation methods mainly include the soil structure stability index (SI), mean weight diameter (MWD), geometric mean diameter (GMD), fractal dimension (D) and the proportion of >0.25 mm aggregates ($R_{0.25}$) [2]. Soil aggregates and their stability are pivotal for soil carbon storage, soil porosity, compactness, water retention, soil water conductivity and erosion resistance [3–6]. A robust soil structure can markedly improve soil microbial community structure, enrich biodiversity, promote nutrient cycling and reduce soil carbon emissions [7,8].

Soil aggregate and its stability are closely related to soil particle composition, SOM, land use and root characteristics [9–11]. Specifically, clay can adsorb more organic matter and humus, resist soil microorganisms decomposing organic carbon and accelerate aggregate formation and stabilization [12]. SOM, the main cementing agent of soil aggregates, is positively correlated with soil aggregate stability [13]. For land use types, there were statistically significant differences in aggregate structure characteristics between forest, shrubland, grassland and cropland [9]. In general, the content of macro-aggregates in forest, shrubland and grassland are higher than that in cropland, so the aggregate stability of cropland is lower [14]. Roots make a major contribution to SOM formation and can enhance soil aggregation through physical entanglement or exudates of cementing

substances [11]. Previous studies on soil aggregates mainly focused on the effects of land use types on aggregates and their stability, including the conversion of other ecosystems to cropland, cropland to artificial grassland or natural grassland, and cropland to economic forest or secondary forest [9,15,16]. However, there are few reports on soil aggregates and their stability when alpine grassland is converted to cropland or abandoned land.

Alpine grassland is the main vegetation type in high-altitude regions [17–19]. Due to the frigid climate, alpine grassland has low soil microbial activity, high root biomass and rich soil organic carbon (SOC) content [20,21], which are closely related to soil aggregates [22,23]. With the development of alpine agriculture, alpine grassland has been converted to cropland and even to abandoned land [17]. So far, studies on soil aggregates in the conversion of grassland to cropland and abandoned land have mainly focused on temperate grassland [24–26]. Concretely, the research of Zhu et al. (2017) on the Loess Plateau found that the SOC content in natural grassland was apparently richer than that in forest, and the carbon addition of grassland could promote the soil aggregate stability [26]. Wang et al. (2018) supported the idea that temperate grasslands that succeeded on abandoned land had higher organic matter and more soil macro-aggregates than cropland [24]. Xiao et al. (2020) suggested that plant roots exert the most significant impact on aggregate stability by studying soil aggregates in temperate grasslands that were converted from abandoned land with different secondary succession gradients [25]. Studies on soil aggregates and their stability due to land use changes from alpine grassland are rarely reported, leading to the fact that the impact mechanism of land use changes from alpine grassland on soil aggregate changes has not been clarified.

The study area lies within the Huangshui River watershed of the Qinghai–Tibet Plateau, where the mean altitude was 2831 m, covering an overall area of $1.61 \times 10^4$ km$^2$, with cropland accounting for 32.50% and the grassland area accounting for 43.73% [27]. Because of the economy's development, a large number of alpine grasslands were reclaimed as cropland, but due to the "Grain for Green Project" and other reasons, a certain amount of cropland was abandoned [28,29]. Therefore, this study selected alpine grassland, cropland and abandoned land as the research objectives and compared and analyzed changes in soil aggregate fractions and stability in order to achieve the following goals: (1) reveal the variation characteristics of soil aggregate composition and stability in the process of land use transformation; (2) explore the relationship between soil aggregate characteristics and soil properties; (3) identify the impact of reclamation and abandonment on soil aggregates in alpine grassland. At the same time, this study can provide data support for the rational utilization of land resources in high-altitude watershed scales and deepen the understanding of soil protection mechanisms, which can provide a theoretical basis for local managers to carry out their work effectively.

## 2. Materials and Methods

### 2.1. Study Area

The study area falls within the Huangshui River watershed (100°42′–103°40′ E, 36°20′–37°28′ N) on the eastern portion of the Qinghai–Tibet Plateau, generally encompassing $1.61 \times 10^4$ km$^2$ (Figure 1). It is a transitional area between the Tibetan and Loess plateaus, with an elevation ranging from 1578 to 4834 m. The annual mean temperature is 0.6–7.9 °C, the annual mean sunshine hours is 2430.8–2666.7 h and the annual precipitation is 30–600 mm. The soil type in the watershed is primarily chestnut soil. Apart from some forests, alpine grassland is the main native vegetation type, accounting for 43.73% of the total watershed area, while cropland is widely distributed on both sides of the river valley, accounting for 32.50% of the study area [27]. The main crops in the watershed are spring wheat (*Triticum aestivum* Linn.), spring maize (*Zea mays* L.), potato (*Solanum tuberosum* L.), spring rape (*Brassica campestris* L.) and spring beans (*Glycine max* (Linn.) *Merr.*). In addition, barley and forage crops are planted in areas with higher altitudes.

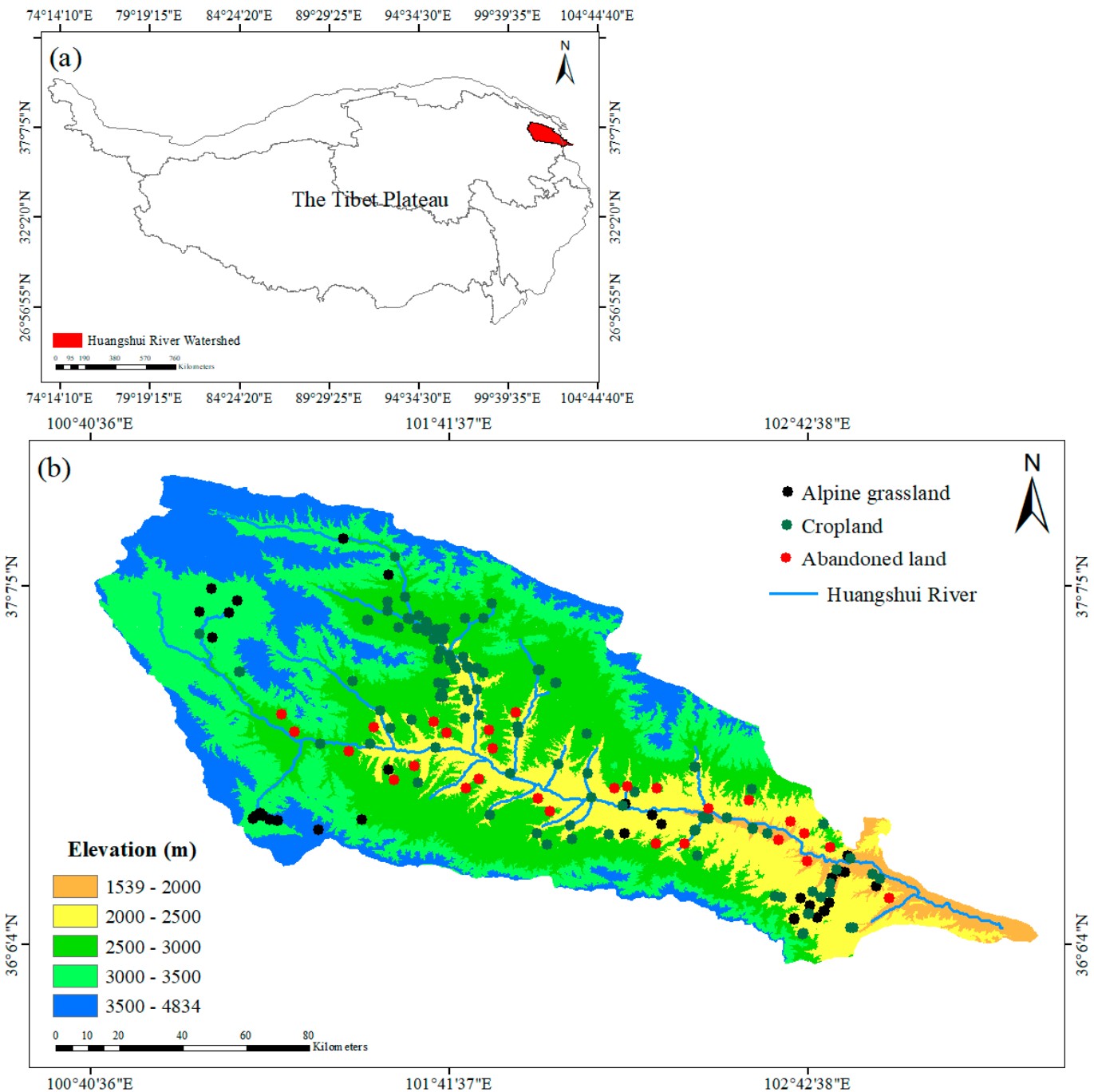

**Figure 1.** The study area location (**a**) and sampling site distribution in Huangshui River watershed (**b**).

### 2.2. Experimental Design and Field Sampling

In April 2021, the sampling sites were ascertained according to the spatial distribution and relative acreage of alpine grasslands, cropland, and abandoned land in the Huangshui River watershed. Because of the high mountains and gullies in the catchment, it was not possible to collect all the sampling points that were set up; a total of 161 sampling sites were finally selected (Figure 1). There are 30 sampling sites of alpine grassland, 94 sampling sites of cropland and 37 sampling sites of abandoned land. At each sampling site, two unperturbed soil blocks (9 cm in diameter and 20 cm in height) were taken from the 0–20 cm and 20–40 cm soil layers, totaling 322 samples. Land use types were surveyed before sampling, and field management surveys were investigated by interviewing local farmers. All soil samples were stored for further analysis after being air-dried in the laboratory.

The air-dried soil samples were screened through a 7 mm sieve for removing large gravel and removing visible pieces of organically grown matter. From each site, 200 g of air-dried soil samples were taken, which were isolated by wet sieving to obtain macro-aggregates (>0.25 mm), micro-aggregates (0.25–0.053 mm) and silt+clay (0.053 mm), then dried in an oven at 45 °C for more than 24 h and weighed [30]. A small amount of air-dried soil sample was collected from each plot and passed through a 1 mm sieve for measuring total SOC content and soil pH. In addition, a certain volume of soil sample (9 cm in diameter, 20 cm in height) was positioned on a 1 mm sieve and washed with running water to obtain fine roots. These roots were oven-dried at 65 °C for 48 h, and weighed to calculate the dry weight density.

The SOC was determined with a mixture of concentrated sulfuric acid and potassium dichromate ($K_2Cr2O_7$–$H_2SO_4$) [31]. The soil pH was measured in a 1:2.5 (soil:water) suspension [32]. Soil particle size composition was determined using a Mastersizer 3000 (Malvern Instruments, Malvern, UK).

### 2.3. Data Analysis

The impacts of land use change on soil stability were evaluated in terms of their MWD, GMD, D, $R_{0.25}$ and SI values. Equations for the first two are as follows:

$$\text{MWD} = \sum_{i=1}^{n} x_i \times w_i \tag{1}$$

$$\text{GMD} = \exp\left[\frac{\sum_{i=1}^{n} w_i \times lnx_i}{\sum_{i=1}^{n} w_i}\right] \tag{2}$$

where $x_i$ is the average diameter of the aggregate class (mm) and $w_i$ is the proportion of each aggregate class in relation to the total aggregate weight.

$$\left[\frac{\overline{d}_i}{d_{max}}\right]^{3-D} = \frac{W(r < \overline{x}_i)}{W_0} \tag{3}$$

where $D$ is the fractal dimension, $\overline{d}_i$ is the mean aggregate diameter (mm) of the ith size class, $d_{max}$ is the average diameter of the largest aggregate, $W(r < \overline{x}_i)$ is the accumulated mass of aggregates of the $i$th size less than $d_{max}$ and $W_0$ is the total mass of the aggregates.

$$\text{SI} = \frac{1.274 \times SOC}{silt + clay} \times 100 \tag{4}$$

where SI is the soil structure stability index, $SOC$ is the soil organic carbon (g kg$^{-1}$), *silt* is the silt content (%) and *clay* is the clay content (%).

### 2.4. Statistical Analysis

One-way ANOVA analysis with a follow-up LSD test was performed to determine whether the soil physicochemical properties, aggregate fraction and soil particle composition differed significantly between land use types. The effects of land use type, soil depth and their interaction on soil properties and aggregate stability were tested using two-way ANOVA. A linear model for multiple factors influencing aggregate stability with land use change was established using stepwise multiple regression. The mean and standard error of each land use type were determined using soil from each sampling location as a replicate. All statistical analyses were performed in IBM SPSS Statistics version 25 and plotted with Origin-Pro 2021.

## 3. Results

### 3.1. Soil Characteristics

The differences in soil pH, root dry weight density (RDWD) and SOC content between alpine grassland, cropland and abandoned land were significant ($p < 0.05$). However, there were no significant differences in median particle size ($D_{50}$) and volume fractal dimension ($D_v$) in different land use types ($p > 0.05$) (Table 1). The soil pH of alpine grassland was significantly lower than that of cropland and abandoned land, and the soil pH of abandoned land was the highest, whether in the 0–20 cm or 20–40 cm layers ($p < 0.05$) (Table 1). At the same time, the RDWD and SOC of alpine grassland were significantly higher than those of cropland and abandoned land ($p < 0.05$), and there was no significant difference in SOC and RDWD between cropland and abandoned land ($p > 0.05$) (Table 1). Two-way ANOVA showed that SOC was significantly affected by land use type, soil depth and their interaction ($p < 0.05$) (Table 2).

**Table 1.** Basic characteristics and soil organic carbon content of different land use types.

| Land Use Type | Soil Depth (cm) | pH | RDWD (g cm$^{-3}$) | SOC (g kg$^{-1}$) | $D_{50}$ (µm) | $D_v$ (mm) |
|---|---|---|---|---|---|---|
| Alpine grassland | 0–20 | 7.86(0.58) c | 0.00171 (0.00077) a | 28.10(16.45) a | 18.14(5.18) a | 2.72(0.02) a |
| | 20–40 | 8.11(0.46) B | 0.00117 (0.00093) A | 18.71(8.30) A | 17.52(4.00) A | 2.71(0.02) A |
| Cropland | 0–20 | 8.11(0.16) b | 0.00051 (0.00037) b | 16.79(6.84) b | 20.73(6.82) a | 2.71(0.02) a |
| | 20–40 | 8.25(0.26) A | 0.00023 (0.00023) B | 13.98(8.20) B | 18.88(4.89) A | 2.71(0.02) A |
| Abandoned land | 0–20 | 8.28(0.10) a | 0.00082 (0.00069) b | 13.66(4.31) b | 17.73(5.56) a | 2.72(0.02) a |
| | 20–40 | 8.34(0.13) A | 0.00013 (0.00006) B | 13.46(4.92) B | 16.99(6.64) A | 2.72(0.03) A |

Notes. The values are the mean and standard deviation. RDWD = root dry weight density; SOC = soil organic content; $D_{50}$ = median particle size; $D_v$ = volume fractal dimension. Different lower and upper cases mean remarkable difference in the same soil layer at different locations ($p < 0.05$).

In the 0–20 cm soil layer, silt and sand were significantly different between cropland and abandoned land ($p < 0.05$) (Figure 2). However, the distribution of clay, silt and sand in the 20–40 cm soil layer showed no remarkable difference ($p > 0.05$) (Figure 2). In general, silt and sand were significantly affected by land use type ($p < 0.05$), though clay was not ($p > 0.05$) (Table 2).

### 3.2. Variation Characteristics of Soil Aggregates

In this study, the SI index was used to evaluate the accuracy of the wet sieving method. MWD, GMD and D were calculated in light of results from each fraction of soil aggregate size. Because the SI was significantly correlated with the soil aggregate stability index, the wet sieving method was deemed accurate for estimating MWD, GMD, D and $R_{0.25}$ (Figure 3).

In the 0–20 cm soil layer, the relative abundance of macro-aggregates (>0.25 mm) was the greatest in alpine grassland, followed by abandoned land, and the least in cropland (Figure 4a). The proportions of macro-aggregates in alpine grassland, abandoned land and cropland were 64.62%, 49.48% and 39.69%, respectively. Macro-aggregates were significantly reduced in proportion when the alpine grassland was converted to cropland ($p < 0.05$), but when the latter was abandoned, macro-aggregates significantly increased ($p < 0.05$). However, abandoned land still harbored a noticeably lower proportion of macro-aggregates than alpine grassland ($p < 0.05$) (Figure 4a). Similar patterns were found in the 20–40 cm soil layer for the relative distribution of macro-aggregates.

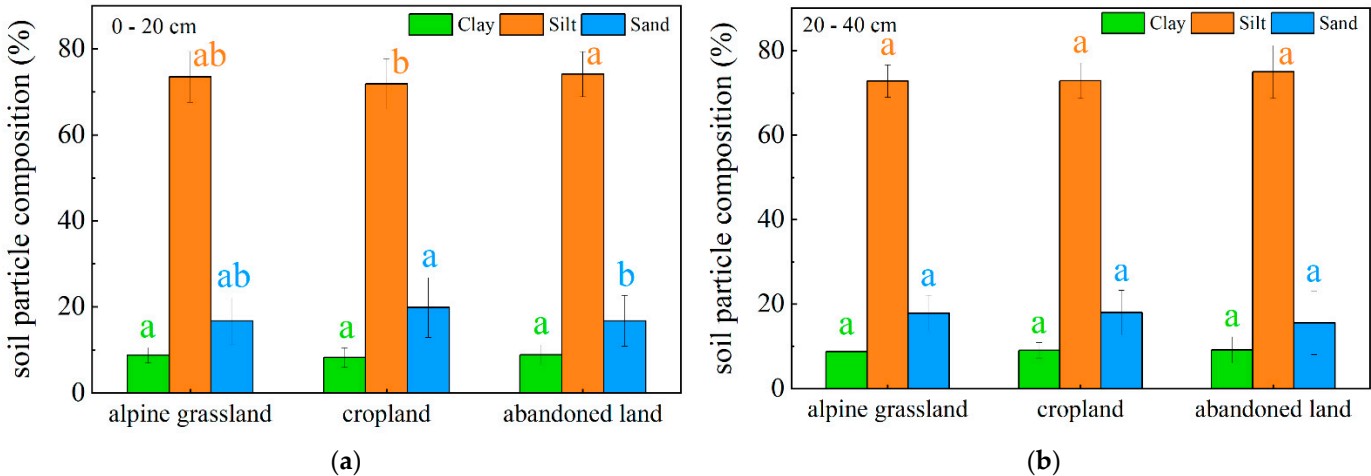

(**a**)                    (**b**)

**Figure 2.** Soil particle composition at 0–20 cm (**a**) and 20–40 cm (**b**) depths of various land use types. Notes. The values are the mean and standard deviation. Different lower case letters indicate significant differences of soil particle composition in the same soil layers by type of land use ($p < 0.05$). The values were compared between bar charts of the same color.

**Table 2.** The impacts of land use type, soil depth and their interaction on soil aggregate stability and properties.

| Factor | Variable | df | F-Value | *p*-Value |
|---|---|---|---|---|
| Land use change | SOC | 2 | 13.967 | 0.000 ** |
| | Clay | 2 | 0.599 | 0.550 |
| | Silt | 2 | 3.814 | 0.023 * |
| | Sand | 2 | 4.557 | 0.011 * |
| Soil depth | SOC | 1 | 12.386 | 0.001 ** |
| | Clay | 1 | 1.214 | 0.272 |
| | Silt | 1 | 0.255 | 0.614 |
| | Sand | 1 | 0.453 | 0.501 |
| Land use change × Soil depth | SOC | 2 | 3.870 | 0.022 * |
| | Clay | 2 | 0.695 | 0.500 |
| | Silt | 2 | 0.343 | 0.710 |
| | Sand | 2 | 0.684 | 0.506 |
| Land use change | MWD | 2 | 28.520 | 0.000 ** |
| | GMD | 2 | 50.826 | 0.000 ** |
| | D | 2 | 40.962 | 0.000 ** |
| | $R_{0.25}$ | 2 | 34.932 | 0.000 ** |
| Soil depth | MWD | 1 | 2.173 | 0.553 |
| | GMD | 1 | 4.410 | 0.142 |
| | D | 1 | 2.079 | 0.037 * |
| | $R_{0.25}$ | 1 | 0.354 | 0.151 |
| Land use change × Soil depth | MWD | 2 | 0.699 | 0.498 |
| | GMD | 2 | 0.312 | 0.732 |
| | D | 2 | 1.588 | 0.206 |
| | $R_{0.25}$ | 2 | 0.161 | 0.851 |

Notes. * Correlation is significant at $p < 0.05$. ** Correlation is significant at $p < 0.01$. SOC = soil organic content; Clay = clay content; Silt = silt content; Sand = sand content; MWD = mean weight diameter; GMD = mean geometric diameter; D = fractal dimension; $R_{0.25}$ = proportion of macro-aggregates.

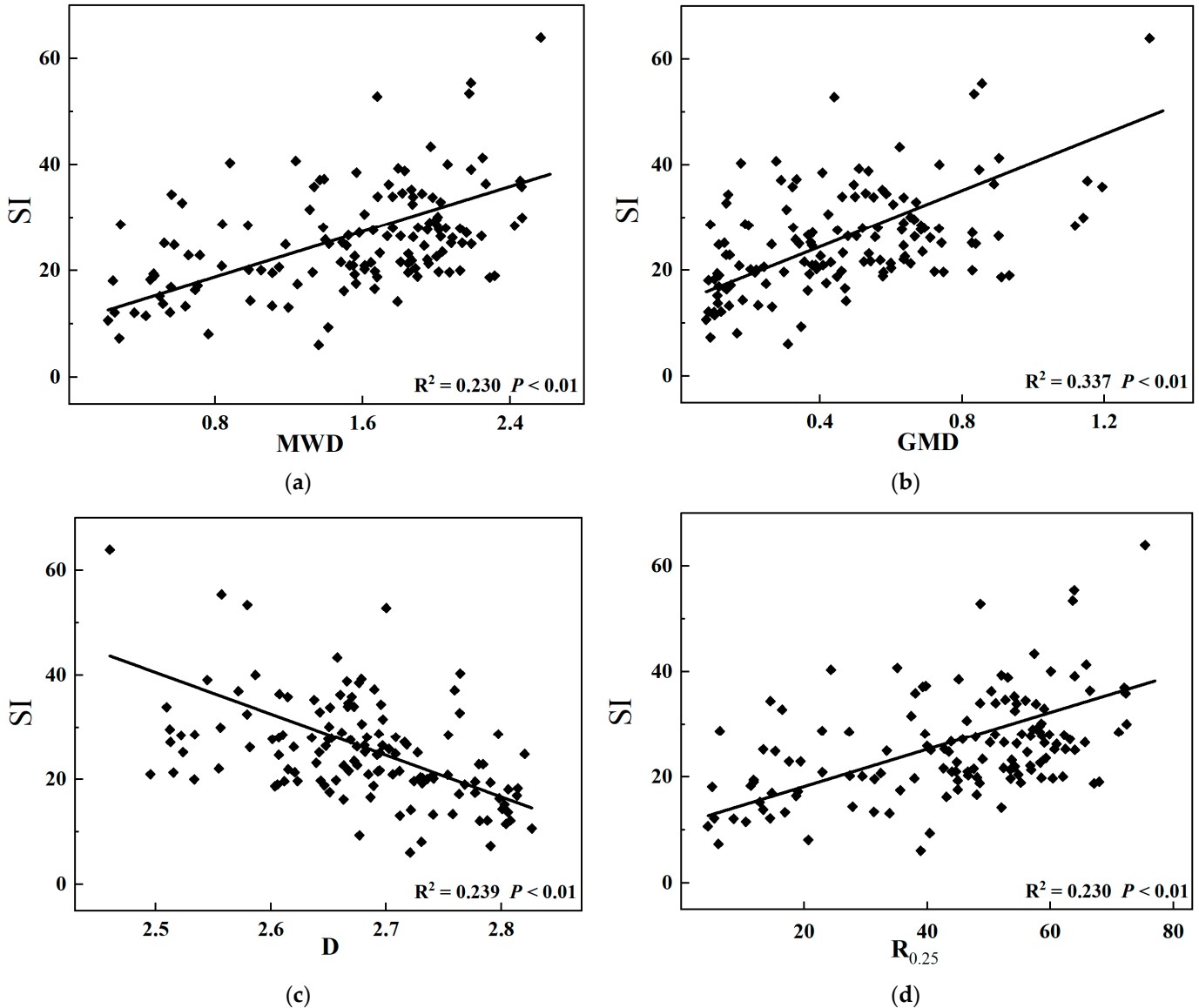

**Figure 3.** Relationships between the soil structure stability index (SI) and four indices of soil aggregate stability at 0–40 cm soil depth. Notes. The subfigures (**a**–**d**) show the relationship between SI and MWD, GMD, D, $R_{0.25}$, respectively.

Micro-aggregates (0.053–0.25 mm) and silt+clay (<0.053 mm) were converted from macro-aggregates after crushing, and had an opposite distribution trend to that of macro-aggregates. The distribution of micro-aggregates and silt+clay in the 0–20 and 20–40 cm soil layers was highest in cropland, followed by abandoned land, and lowest in alpine grassland. In the 0–20 cm soil layer, there was no remarkable difference in the distribution of micro-aggregates between cropland and abandoned land ($p > 0.05$). In the 20–40 cm soil layer, the proportion of silt+clay in the abandoned land was slightly higher than that in the alpine grassland, but the difference was not significant ($p > 0.05$) (Figure 4a,b).

Figure 5 shows the stability indices of soil aggregates under various land use types. In the 0–20 cm layer, the MWD differed typically between the three land use types, being highest in alpine grassland and lowest in cropland ($p < 0.05$) (Figure 5a). The values of GMD and $R_{0.25}$ exhibited patterns closely resembling that of MWD and were also characterized by significant differences between the land use types ($p < 0.05$) (Figure 5b,d). The D values in the entire soil layers differed significantly between land use types, with alpine grassland having the lowest, abandoned land having the second lowest, and cropland having the

highest ($p < 0.05$) (Figure 5c). In summary, the soil aggregate stability under different land use types decreased in the order of alpine grassland > abandoned land > cropland.

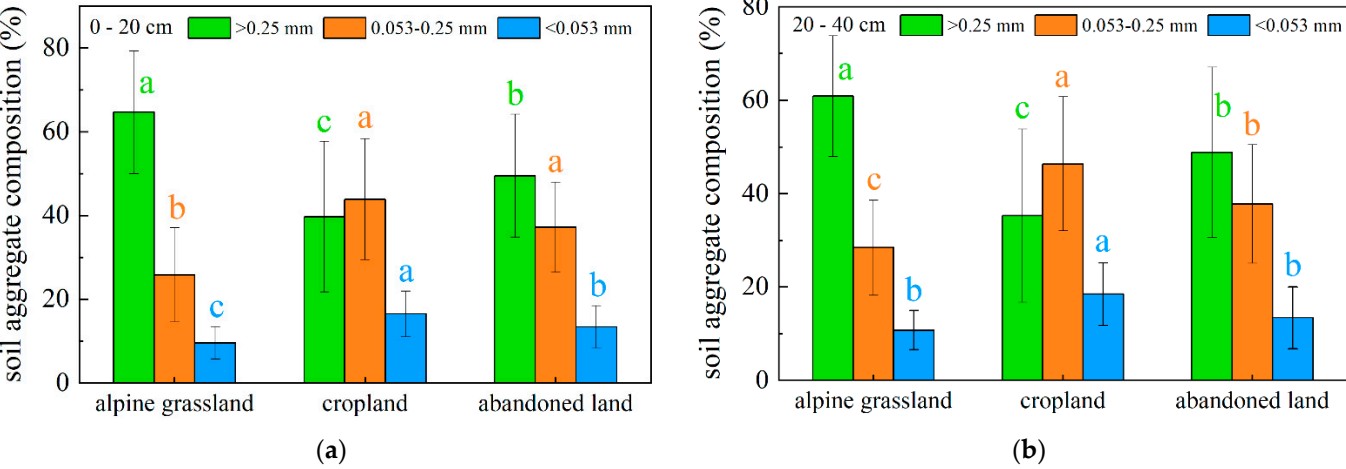

**Figure 4.** Soil aggregate composition at 0–20 cm (**a**) and 20–40 cm (**b**) depths of various land use types. Notes. The values are the mean and standard deviation. Different lower case letters indicate significant differences of soil aggregate composition in the same soil layers by the type of land use ($p < 0.05$). The values were compared between bar charts of the same color.

### 3.3. Correlations of Aggregate Stability with SOC and Soil Particle Composition

Soil aggregate stability was significantly affected by land use type ($p < 0.01$) (Table 2). In both the 0–20 and 20–40 cm soil layers, the stability indices of soil aggregates (except for D) were ranked alpine grassland > abandoned land > cropland ($p < 0.05$) (Figure 5). MWD, GMD and $R_{0.25}$ were all noticeably positively correlated with SOC content, whereas D was significantly negatively correlated and soil aggregate stability was positively correlated with SOC in the 0–20 and 20–40 cm soil layers ($p < 0.01$) (Figures 6 and 7) (Table 3).

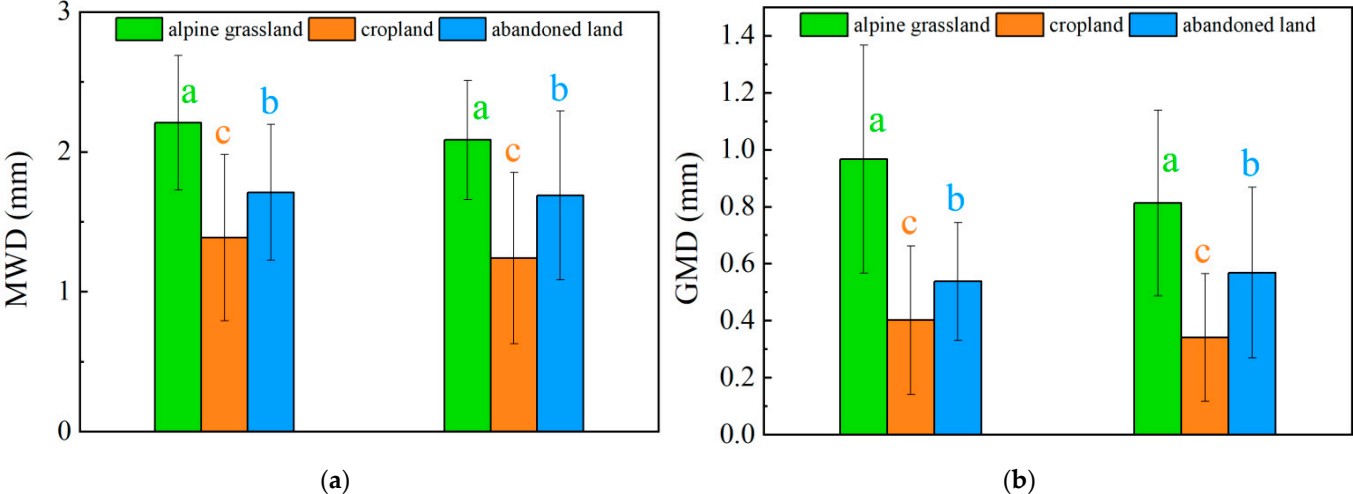

**Figure 5.** *Cont.*

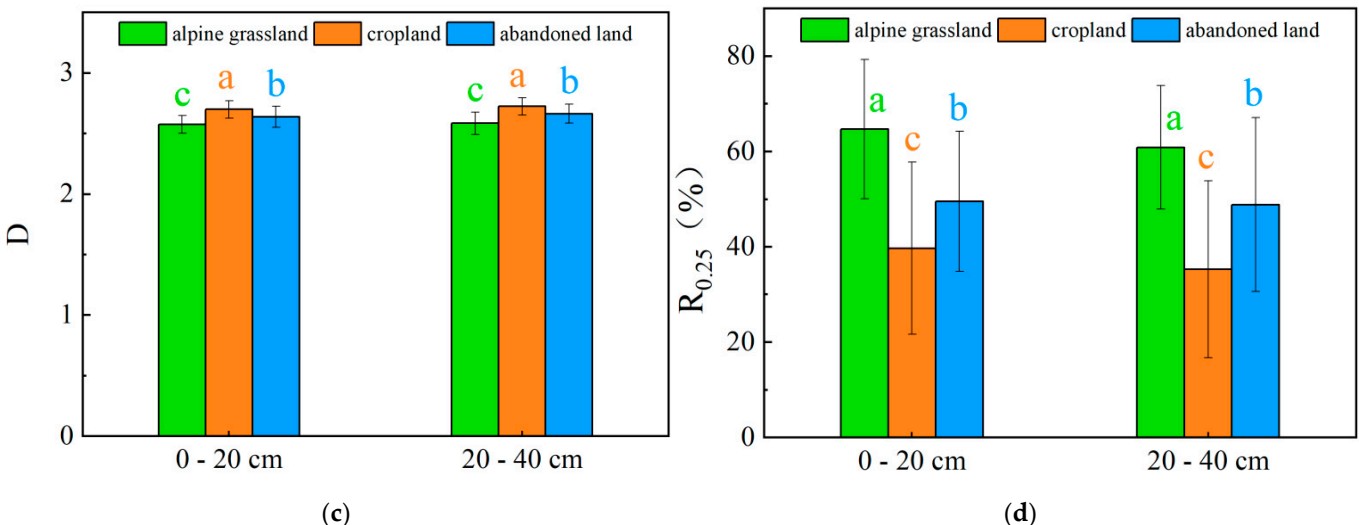

**(c)**                            **(d)**

**Figure 5.** Variations of soil aggregate stability indices. Notes. The values are the mean and standard deviation. Different lower case letters indicate significant differences of soil aggregate stability in the same soil layers at the different locations ($p < 0.05$). The values were compared between bar charts of the same color. The subfigures (**a**–**d**) show the soil aggregate stability of 0–20 cm and 20–40 cm depths for MWD, GMD, D, and $R_{0.25}$ under different land use types, respectively.

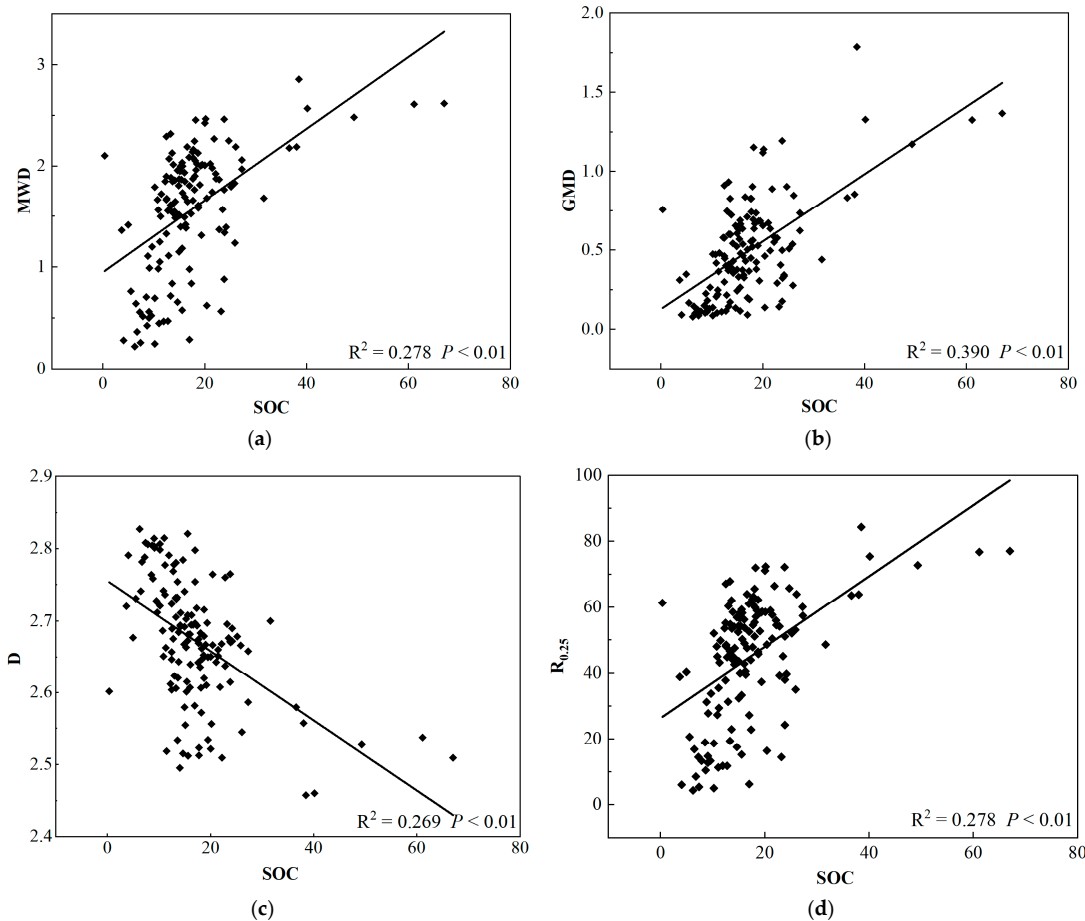

**Figure 6.** The soil aggregate stability indices as a function of the SOC at 0–20 cm soil depth. Notes. The subfigures (**a**–**d**) show the relationship between SOC and MWD, GMD, D, $R_{0.25}$ at 0–20 cm soil depth, respectively.

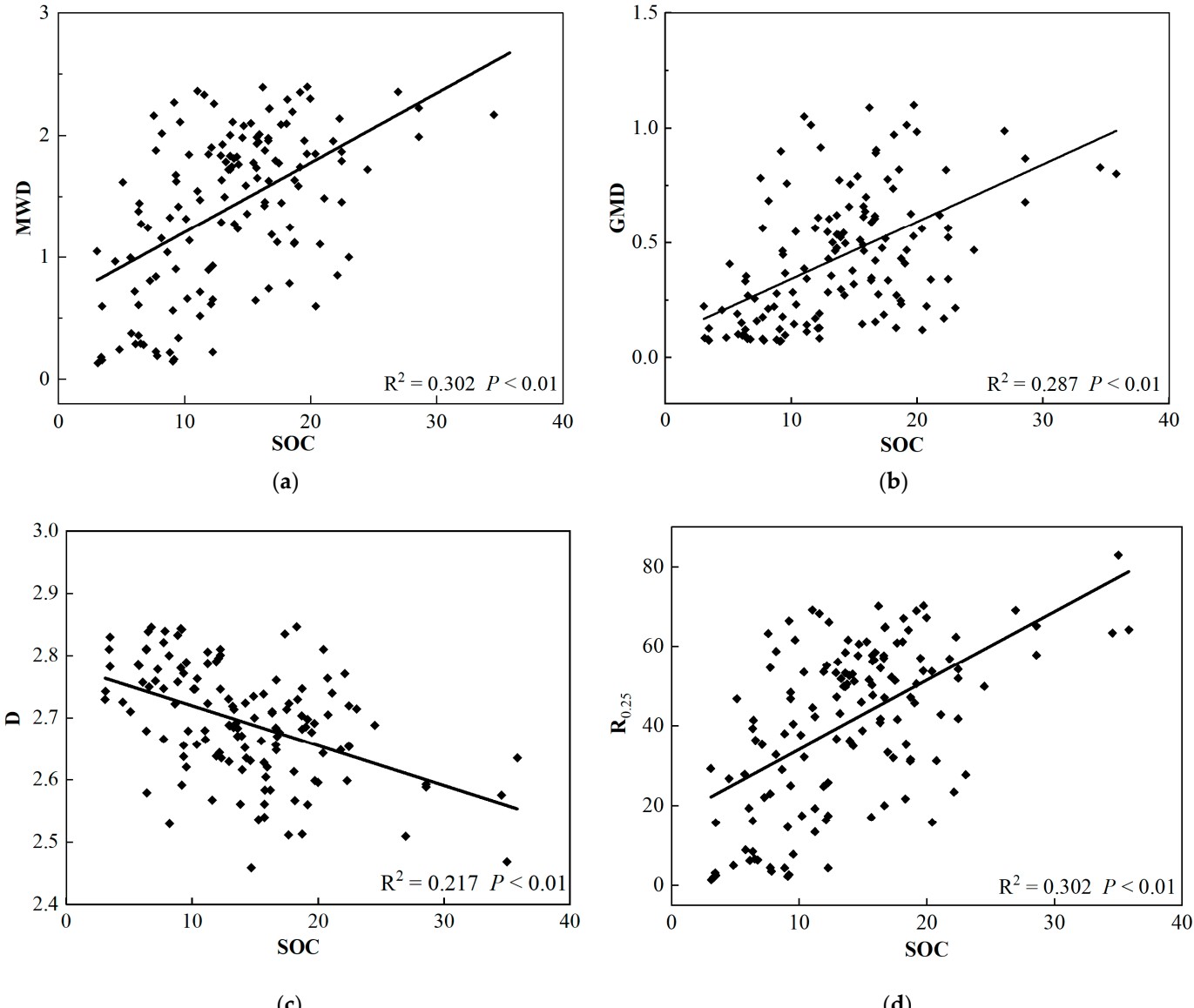

**Figure 7.** The soil aggregate stability indices as a function of the SOC at 20–40 cm soil depth. Notes. The subfigures (**a**–**d**) show the relationship between SOC and MWD, GMD, D, $R_{0.25}$ at 20–40 cm soil depth, respectively.

In the 0–20 cm soil layer, clay (<0.002 mm) and silt (0.002–0.05 mm) content were found to be significantly positively correlated with soil aggregate stability, and their $R^2$ was 0.127–0.179 ($p < 0.05$) (Figure 8). In contrast, sand (0.05–2 mm) content was negatively correlated with soil aggregate stability ($p < 0.01$), and its $R^2$ was 0.229–0.283 (Figure 8). In the 20–40 cm soil layer, there was no significant correlation between clay content and aggregate stability indices ($p > 0.05$) (Figure 9). The relationship between the silt and sand content and aggregate stability was the same as in the 0–20 cm layer, and the $R^2$ of these were 0.112–0.226 and 0.129–0.230, respectively (Figure 9). Additionally, sand was more significant in affecting soil aggregate stability in this study ($p < 0.05$) (Figures 8 and 9) (Table 3).

**Table 3.** Multiple regression analysis of soil aggregate stability indices and soil properties.

| Soil Depth (cm) | Dependent Variable | Formula | $R^2$ |
|---|---|---|---|
| 0–20 | MWD | $MWD = 1.503 + 0.031 \times SOC - 0.031 \times D_{50}$ | 0.446 ** |
| | GMD | $GMD = 0.288 + 0.017 \times SOC - 0.009 \times D_{50}$ | 0.529 ** |
| | D | $D = 2.781 - 0.005 \times SOC$ | 0.401 ** |
| | $R_{0.25}$ | $R_{0.25} = 43.288 + 0.932 \times SOC - 0.949 \times D_{50}$ | 0.446 ** |
| 20–40 | MWD | $MWD = 56.234 + 0.024 \times SOC - 0.162 \times D_{50} - 17.712 \times D_v - 0.058 \times Silt$ | 0.492 ** |
| | GMD | $GMD = 0.320 + 0.014 \times SOC - 0.009 \times D_{50}$ | 0.459 ** |
| | D | $D = -4.442 - 0.002 \times SOC - 0.021 \times D_{50} + 2.278 \times D_v + 0.008 \times Silt$ | 0.377 ** |
| | $R_{0.25}$ | $R_{0.25} = 1701.850 + 0.715 \times SOC - 4.898 \times D_{50} - 536.971 \times D_v - 1.178 \times Silt$ | 0.493 ** |
| 0–40 | MWD | $MWD = -2.508 + 0.030 \times SOC - 0.095 \times D_{50} + 0.096 \times Sand + 0.047 \times Silt$ | 0.472 ** |
| | GMD | $GMD = 0.294 + 0.014 \times SOC - 0.009 \times D_{50} + 96.033 \times RDWD$ | 0.544 ** |
| | D | $D = 2.717 - 0.003 \times SOC - 0.003 \times D_{50} - 24.106 \times RDWD$ | 0.375 ** |
| | $R_{0.25}$ | $R_{0.25} = -78.937 + 0.907 \times SOC - 2.897 \times D_{50} + 2.913 \times Sand + 1.442 \times Silt$ | 0.473 ** |

Notes. * Correlation is significant at $p < 0.05$. ** Correlation is significant at $p < 0.01$. The values of the included variables are significant at $p < 0.05$. SOC = soil organic content; Clay = clay content; Silt = silt content; Sand = sand content; MWD = mean weight diameter; GMD = mean geometric diameter; D = fractal dimension; $R_{0.25}$ = water-stable macro-aggregate content; RDWD = root dry weight density; $D_{50}$ = median particle size; $D_v$ = volume fractal dimension.

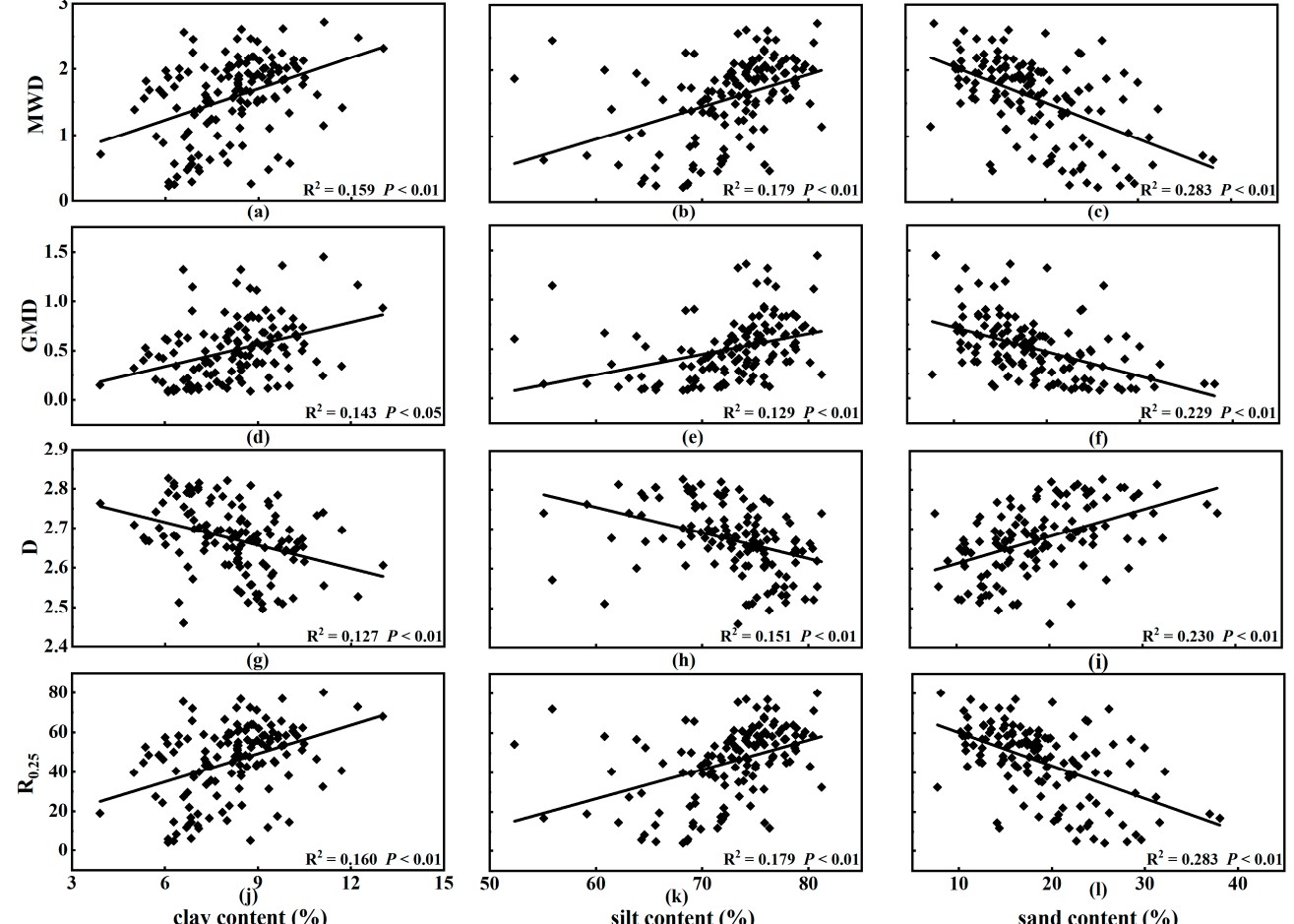

**Figure 8.** The soil aggregate stability indices as a function of the soil particle composition at 0–20 cm depth. Notes. The subfigures (**a–c**) show the relationship between MWD and the soil particle composition; (**d–f**) show the relationship between GMD and the soil particle composition; (**g–i**) show the relationship between D and the soil particle composition; (**j–l**) show the relationship between $R_{0.25}$ and the soil particle composition.

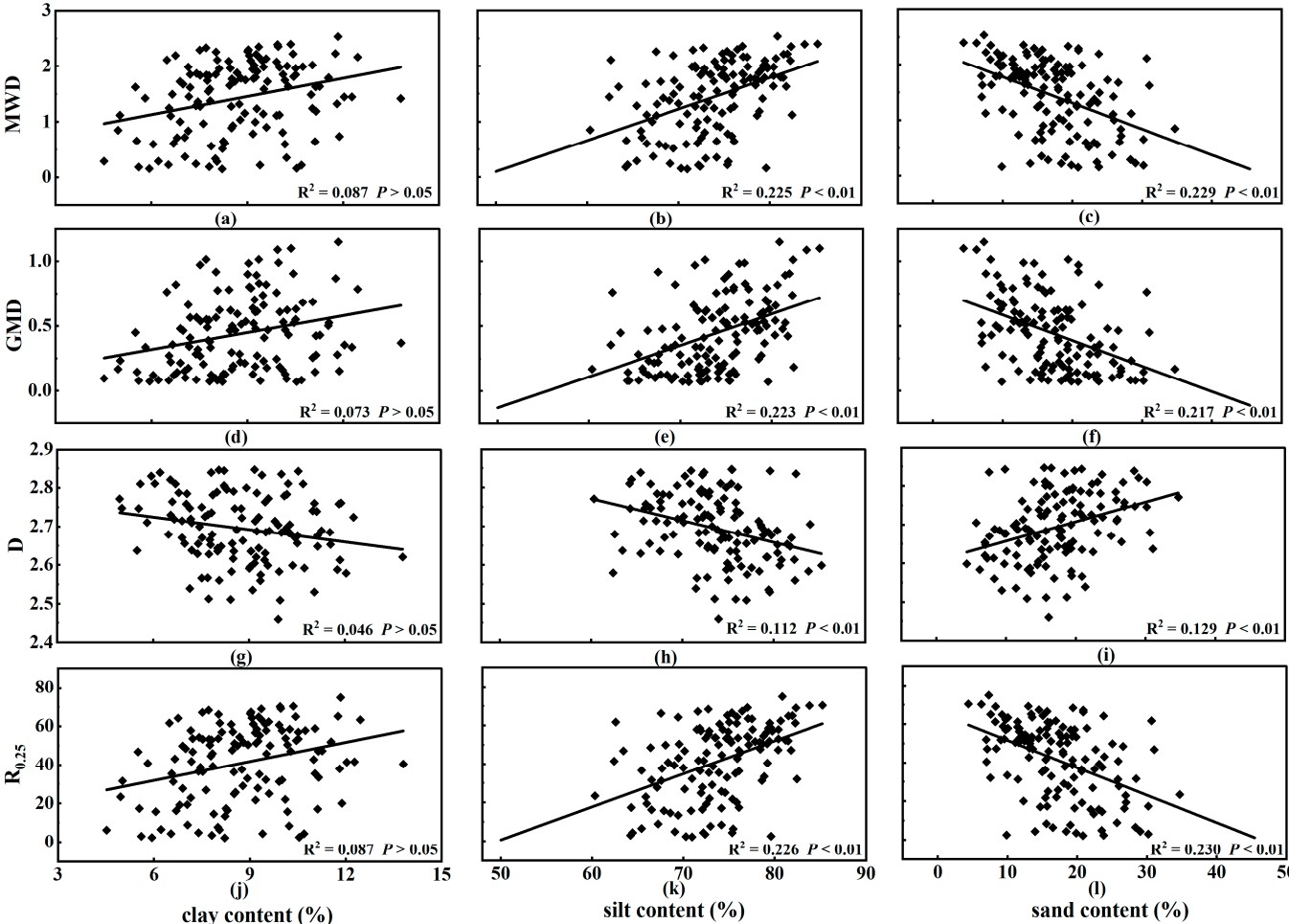

**Figure 9.** The soil aggregate stability indices as a function of the soil particle composition at 20–40 cm depth. Notes. The subfigures (**a**–**c**) show the relationship between MWD and the soil particle composition; (**d**–**f**) show the relationship between GMD and the soil particle composition; (**g**–**i**) show the relationship between D and the soil particle composition; (**j**–**l**) show the relationship between $R_{0.25}$ and the soil particle composition.

## 4. Discussion

### 4.1. Effects of Land Use Type, Soil Depth and How They Interact on Soil Properties

SOC was significantly affected by land use type, soil depth and their interaction. SOC content decreased significantly after alpine grassland was converted to cropland and abandoned land. This result is consistent with the study of Zhu et al. (2021) in the Nangou watershed of the Loess Plateau [16]. The reasons could be attributed to the following: (1) Due to the high groundcover and extensive root system in grassland, more biomass was returned to the soil, which effectively promotes the accumulation of SOM (Table 1) [33]. (2) Crop straws in cropland rarely return to the soil, and crop growth will consume a large amount of SOM, inducing a decrease in the accumulation of organic matter in the soil [34,35]. In this study, no obvious difference in SOC content was found between cropland and abandoned land, which is in line with the result of Liu and Han. (2020). The reason may be that the restoration years of abandoned land are short, which means it cannot provide sufficient organic matter sources for the soil [24,36]. Reclamation of alpine grassland damaged soil aggregate structure, aggravated soil carbon loss and led to soil quality decline. The protection and management of alpine grassland should be strengthened on the existing basis. The scientific use of cropland resources for long-term abandoned land manual management effectively promotes land quality recovery and accelerates the improvement of soil structure.

Silt and sand were significantly affected by land use type. This finding is consistent with the study of Zhu et al. (2017) [26]. This may be due to the soil particle composition, with silt and sand content accounting for the majority of the proportion, being more easily affected when land use changed. However, Liu et al. (2019) found that soil particles were not impacted by land use type [37], likely because of the short time frame of land use change, and it was difficult to have a significant indigenous impact on soil particle composition.

Compared with temperate grassland, SOC content in alpine grassland was higher in this study. Dou et al. (2020) reported that the SOC content of temperate grassland in the 0–20 and 20–40 cm soil layers (7.52 g kg$^{-1}$,4.56 g kg$^{-1}$; 12.70 g kg$^{-1}$, 5.05 g kg$^{-1}$) [9,38] was much lower than that of the two soil layers in this study (28.10 g kg$^{-1}$,18.71 g kg$^{-1}$) (Table 1). The above findings could be explained by the fact that low soil temperature in alpine grasslands inhibits microbial activity and reduces the microbial decomposition of organic matter, thus effectively preventing the loss of organic carbon [39]. In addition, the alpine grassland accumulated huge root biomass, providing a rich source of organic matter for the soil [40].

### 4.2. Distributions of Soil Aggregate Stability

Soil aggregate stability was significantly affected by land use type. This is consistent with the findings of Baranian Kabir et al. (2017), who discovered that soil aggregate stability was highest in alpine grassland and lowest in cropland [41]. On the one hand, grassland and abandoned land had more organic matter input, which can improve microbial activity, promote polysaccharide and cementing agent formation, and enhance the stability of soil aggregates [42–44]. On the other hand, long-term tillage will disturb the macro-aggregates in the soil, resulting in a decrease in soil aggregate stability [45].

In comparison with temperate grassland, soil aggregate stability in alpine grassland was higher in this study. In the 0–20 and 20–40 cm soil layers, the ratio of R$_{0.25}$ (60.84–64.62%) in alpine grassland was much higher than that of soil macro-aggregates (20–30%) in Dou et al. (2020) [9]. The reason for the high proportion of macro-aggregates in alpine grassland may be due to the large root biomass accumulated in alpine grassland and the high SOM content, which can be conducive to macro-aggregate formation in soil (Table 1) [46]. Xiao et al. (2020) found MWD (1.12–1.30 mm) in the 0–20 cm soil layer of temperate grassland that was lower than MWD (1.73–2.69 mm) in this study [25]. Otherwise, a large number of plant roots in alpine grasslands can enhance soil aggregation and promote the formation and stability of aggregates by physical entanglement or the secretion of cementing substances [11,47].

### 4.3. Correlations of Aggregate Stability with SOC and Soil Particle Composition

In this study, SOC was significantly positively correlated with soil aggregate stability. This result is consistent with studies conducted by Liu et al. (2019) [35], and Zhu et al. (2021) [16]. SOC, as the cementing material of aggregates, can effectively increase the number of macro-aggregates, ameliorate aggregate stability and improve soil structure [48]. Moreover, SOC can improve soil aggregate stability by boosting soil hydrophobicity to reduce the aggregates' destruction by rainwater [49].

Sand content was significantly negatively correlated with the stability of soil aggregates in the 0–40 cm soil layer, indicating that sand was more significant in affecting soil aggregate stability in this study. This result is consistent with Barberis et al. (1991). Sand was very important for the formation of macro-aggregates, and it was the main component of macro-aggregates [50]. Specifically, sand and micro-aggregates are combined to form macro-aggregates through biological methods such as fine root and fungal entanglement [50,51], but their aggregate stability is poor and vulnerable to external interference [52].

### 5. Conclusions

In this study, we selected alpine grassland, cropland and abandoned land in the Huangshui River watershed of the Qinghai–Tibet Plateau as the research objects to measure

and analyze the soil aggregate distribution, aggregate stability and influencing factors of 161 soil samples in 0–20 and 20–40 cm soil layers. It was found that the effects of land use changes on soil aggregate characteristics in high-altitude watersheds mainly include the following aspects:

(1) Following cropland abandonment, macro-aggregate distribution and aggregate stability increased, while micro-aggregate and silt+clay decreased significantly. This indicates that tillage will destroy the macro-aggregates in the soil, resulting in decreased aggregate stability; abandoning cropland reduced the destruction of aggregates and improve aggregate stability, but there was still a certain disparity compared with natural grassland.

(2) During the conversion of alpine grassland to cropland and abandoned land, SOC and soil particle composition had significant effects on soil aggregate stability. Among them, SOC content, silt content and clay content were positively correlated with aggregate stability, while sand content was negatively correlated with soil aggregate stability. Clay and silt promoted the formation of macro-aggregates and enhanced aggregate stability, while sand was not conducive to the formation of macro-aggregates and aggregate stability. Additionally, in this study, sand was more likely to affect soil aggregates and their stability.

(3) Compared with temperate grassland, alpine grassland had higher SOC content, soil aggregate stability and distribution of macro-aggregates. Low soil temperature in alpine grasslands inhibits microbial activity, which can effectively prevent the loss of organic carbon. In addition, the alpine grassland accumulated huge root biomass, providing a rich source of organic matter for the soil. However, this result is not absolute, depending on the specific sampling environment. Microclimates may produce different results.

**Author Contributions:** Conceptualization, Y.L. (Ying Li) and Y.Z.; Methodology, Y.L. and Y.Z.; Software, Y.L. (Ying Li); Validation, Y.Z.; Formal analysis, Y.L. (Ying Li); Investigation, Y.L., Z.M., Y.L. (Yutao Liu), C.Z., Q.M. and Z.C.; Resources, Y.Z. and H.S.; data curation, Y.L. (Ying Li); Writing—original draft preparation, Y.L. (Ying Li); Writing—review and editing, Y.L. (Ying Li) and Y.Z.; Visualization, Y.L. (Ying Li); Supervision, Y.Z. and W.W.; Project administration, Y.Z. and H.S.; Funding acquisition, Y.Z. and H.S. All authors have read and agreed to the published version of the manuscript.

**Funding:** This research was funded by the Natural Science Foundation of Qinghai Province, China (grant number 2020-ZJ-967Q), the Thousand High Innovative Talents Program of Qinghai Province (2019) and the National Natural Science Foundation of China (grant number U20A20115, 42207375).

**Institutional Review Board Statement:** Not applicable.

**Informed Consent Statement:** Not applicable.

**Data Availability Statement:** Not applicable.

**Acknowledgments:** The authors are grateful to the editor and reviewers for their valuable comments and suggestions.

**Conflicts of Interest:** The authors declare no conflict of interest.

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
