# Peer review of "Variation in Soil Aggregate Stability Due to Land Use Changes from Alpine Grassland in a High-Altitude Watershed"

_land, doi:10.3390/land12020393_

Round 1

Reviewer 1 Report

Title: Variation in soil aggregate stability due to land use changes 2 from alpine grassland in a high-altitude watershed

Ms No: land-2153946

To

The editor

Journal of land.

Thanks for giving me an opportunity to review this manuscript. I found that authors have done good study in new field of research. I appreciate the efforts of the authors for using good methodology to access the data results. Each parameter (abstract, introduction, material methods, results and discussion including conclusion) of the manuscript described well. I recommend the study to be published in the Journal. I have noticed few points for its improvement for quality publication and its wide readership.

Comments:

·         Please provide pointwise objectives of the study for clear understanding.

·         Line number 87,  2. Materials and methods should be given in new para.

·         Line number 96 the sentence “The soil type in the watershed is chiefly chestnut soil”: is repeated.

·         Line number 100: wheat, spring maize, potatoes, spring rape, and spring beans. Name should be given scientific for wide understanding.

·         I also suggest authors should also read manuscript once again for clear language and other minors errors.

Author Response

Dear reviewer:

Thank you for your decision and constructive comments on my manuscript. We have carefully considered the suggestion and make some changes. We have tried our best to improve and made some changes in the manuscript.

Reviewer 2 Report

The paper is about soil aggregate stability conditions resulting from different land use changes. The paper is in general well structured, although I found results in the Discussion section that have to be moved to Results section. The importance of the study is not clearly stated in the Introduction. The Discussion must be improved by adding insight on the extent, limitation, application and implications of the results for soil management.

Although is well written, the paper would benefit from further English proof reading.

Introduction

Lines 47-49 (and elsewhere). Avoid the use of “best” or “better” or “worse” when talking about statistical results. For instance, “… had the best aggregate stability”, best for what? Instead, use “high” aggregate stability or “low” aggregate stability.

The scope and the importance and relevance of the study should be clearly explained. There is little information regarding why this study is important.

Methods

Line 87. Separate section 2 to line 88.

Figure 2. In my opinion, Figure 2 should be in the Results section even if it represents the evaluation of the accuracy of the wet sieving method. Please consider moving it to Results.

Results

Table 1. It should be stated what the lower and upper cases indicate in terms of statistical comparisons. Tables in Discussion should be moved to Results.

Discussion:

There is no need to present again p values or references to tables and figures presented in the Results. They have already been presented in the Results section. As it is, it seems that results are being repeated in the Discussion section.

Line 200. Avoid using terms such as “good”; good for what? Instead, use high or low.

Tables 2 and 3 must be in the Results section, not in the Discussion.

Implications for soil management are lacking in the Discussion. It seems the authors limited their Discussion to contrast their results with other studies (which is correct), but they don´t show the importance, utility and implications of their results. Authors must show the extent and limitation of application of their findings and the implications of the different land use change transitions on the soil properties studied (for instance, what is the impact of reclamation and abandonment on soil aggregates in alpine grassland, as stated in the objectives?).

There is no need to state p-values in the Conclusions. It would be useful to provide some insight on the implication of the results in the conclusions.

Author Response

(The authors gave the same response as above.)

Reviewer 3 Report

The topic of the article is interesting, especially the research site. However, I have comments on this, especially in a methodological sense.

Soil type is mentioned twice ?  ”The soil type in the watershed is primarily chestnut  soil. The soil type in the watershed is chiefly chestnut soil.“ Please, correct this information.

Samples are „baked“ (line 121) or dried, or heated? Unusual is the use of the term "baked" in the laboratory.

For the determination of SOC using potassium bichromate, further detailing is required, link to the method used. What is the procedure of the analysis, temperature, photometry or titration, the final procedure for determining SOC? What are the weights of the sample for analysis, how were the samples prepared for an analysis?

How is RDWD was determined? Please, describe.

Particle sizes distribution. Particle or particles? Fig.3?

The term “remarkable differences”, I think, is better to use “significant differences”?

Discussion section. Why tables including results are presented in this sections?

Author Response

(The authors gave the same response as above.)

Round 2

Reviewer 2 Report

I have read the manuscript again, and I find that the authors have attended my suggestions and carefully have responded to my comments and amended the manuscript. I am satisfied with this new version and recommend its publication in its present form.